# Effects of Multimodal Rehabilitation on the Activities of Daily Living, Quality of Life, and Burden of Care for Patients with Parkinson’s Disease: A Randomized Control Study

**DOI:** 10.3390/healthcare10101888

**Published:** 2022-09-27

**Authors:** Hyun-Se Choi, Seung-Hyun Cho

**Affiliations:** 1Department of Rehabilitation Medicine, Seoul National University Bundang Hospital, Seongnam 13620, Korea; 2Department of Occupational Therapy, College of Health Sciences and Social Welfare, Woosuk University, Wanju 55338, Korea

**Keywords:** multimodal rehabilitation, Parkinson’s disease, elderly, ADL, QoL, burden

## Abstract

Background: Parkinson’s disease reduces patients’ function, activities of daily living, and quality of life, and increases their guardians’ burden of care. This study verified the effectiveness of a multimodal rehabilitation programme for patients with Parkinson’s disease. Trial design: This study was a multicentre parallel randomised controlled, single-blind, trial conducted in three hospitals in Korea. Methods: A central randomisation centre used computer generated tables to randomly allocate 60 of 75 patients with Parkinson’s disease who fulfilled the study requirements into experimental (*n* = 30; multimodal rehabilitation; consisting of daily living training, guardian education, home environment modification, fine muscle exercise, balance training, and training using auxiliary tools performed 50 min per session, twice a week, in 10 sessions) and control (*n* = 30; traditional rehabilitation; consisting of task-oriented training, joint exercise, and daily living training performed 50 min per session, twice a week, in 10 sessions) groups. Results: Multimodal rehabilitation for Parkinson’s disease significantly improved the activities of daily living (*p* < 0.01) and quality of life of patients (*p* < 0.001) and eased the guardians’ burden of care (*p* < 0.001). Conclusions: Multimodal rehabilitation is suggested to improve activities of daily living, quality of life of patients with Parkinson’s disease, and reduce the burden of care of their guardians.

## 1. Introduction

Parkinson’s disease is a neurodegenerative disease caused by the effects of motor and non-motor brain structures, which leads to various problems such as impairment of gait, balance, hand agility, memory, or executive function [1,2]. Parkinson’s disease is the second most common neurodegenerative disease after dementia, with a global prevalence of approximately 1–2% among older adults aged ≥ 65 years [3]. The intelligence level of patients with Parkinson’s disease is relatively normal. However, as the disease progresses, it is characterized by typical clinical symptoms, such as bradykinesia, rigidity, resting tremor, and postural change [4].

For patients diagnosed with Parkinson’s disease, the primary problem is performing basic activities of daily living, such as personal hygiene, dressing, and using the bathroom [5]. Activities of daily living are greatly affected by the disruption in motor abilities due to Parkinson’s disease, of which, bradykinesia as compared with tremors and stiffness, has a considerable effect on patients’ activities of daily living [6]. By five years after diagnosis, the percentage of patients’ who depend upon others to accomplish activities of daily living is approximately 10–25% and by 10 years, 20–50%. This extent of dependence is more when patients are classified based on the Hoehn and Yahr scale as stage 3 or higher [7].

In the early stages of Parkinson’s disease, the perceived quality of life of the patients is not substantially affected because the condition is controlled through the use of drugs; however, quality of life can diminish over time due to manifestations of motor symptoms (e.g., restriction of activity) and non-motor symptoms (e.g., depression and fatigue) [8]. Personal hygiene of Parkinson’s disease patients and freezing during ambulation when performing activities of daily living, are the main factors that decrease the quality of life [9]. In particular, the quality of life of patients with Parkinson’s disease is approximately 14% lower than that of stroke patients, and patients with Parkinson’s disease complain of more pain than patients suffering from other neurodegenerative diseases, such as amyotrophic lateral sclerosis and multiple sclerosis. The quality of life is reported to be lower still as it is difficult for doctors to detect pain in patients with Parkinson’s disease [10,11].

Guardians of patients with Parkinson’s disease have many responsibilities, such as coordinating care, administering medications, aiding in personal hygiene, communication, advocacy, and fall prevention, as well as providing emotional support. As the disease progresses, their role of directly supporting activities of daily living increases [12]. Moreover, more than 25% of these guardians spend approximately 70 h a week caring for patients with Parkinson’s disease, which often requires them to reduce their working hours or give up their jobs [13]. Consequently, guardians of patients with Parkinson’s disease are negatively affected by the burden of support in terms of physical, mental, and economic aspects [14]. Guardians of patients with Parkinson’s disease are mainly spouses, and they complain of a relatively higher level of burden of care than those caring for patients with strokes or general chronic diseases. The level of burden of care is higher and a higher stage in the Hoehn and Yahr stage scale is often observed [15]. As reported, improvement in the patients’ level of concentration and reduction in the guardian’s level of depression affect the guardian’s burden of care, and that providing parental education is effective in reducing the burden of care [16].

Rehabilitation therapy is a non-drug treatment strategy and has been proven effective in alleviating symptoms of Parkinson’s disease [17]. It is considered as an adjunct to drug and surgical treatment for Parkinson’s disease to maximise functional improvement and minimise secondary complications [18]. The objectives of rehabilitation for patients with Parkinson’s disease include maintaining or improving functional mobility and walking ability; improving personal hygiene management ability; improving motor function related to safety (fall prevention); modifying the home environment; and improving upper extremity function [19].

Several studies have attempted to improve the activities of daily living, quality of life, and burden of the guardians of patients with Parkinson’s disease. Hand agility training, paper, and pencil exercise, writing training using visual/verbal strategies, and training for activities of daily living using self-management and cognitive behavioural strategies for patients with Parkinson’s disease have been employed [20]. Goal-based exercise, daily management learning, and strategies for enhancing individual autonomy, have been employed for patients with Parkinson’s disease [21]. Patients with Parkinson’s disease are trained to engage in leisure activities, such as modifying their home environment for grooming and functional mobility, working (unpaid or paid), shopping, and watching movies; meanwhile, their guardians are trained on how to supervise and support their activities of daily living. The goals are to encourage the patients to participate in meaningful activities and roles and to create an environment for self-management [22]. A previous study implemented a reward strategy for improving task performance, simplification of activities, habituation, and environmental modification, and applied auxiliary tools to improve independence, safety, and efficiency for patients with Parkinson’s disease [23]. The guardians were provided with information on Parkinson’s disease, introduced to assistive devices for performing activities of daily living, and trained in supervision skills. Prescribing physical activity to patients with Parkinson’s disease should allow them to participate in meaningful exercise, use cues in performing activities of daily living, and consider cognitive impairment or depression related to Parkinson’s disease, while exploring the application of patient-centred self-management strategies to enhance self-efficacy, environmental modification, and community resources for social interaction [24]. These studies have set goals to improve performance and independence in patients with Parkinson’s disease using multimodal methods such as functional training, training for activities of daily living, environmental modification, and guardian education. Thus, in this study, we aimed to assess whether the application of multimodal rehabilitation among older adults with Parkinson’s disease and their guardians can improve daily life, quality of life, as well as ease the burden of support. It is hoped that Parkinson’s disease can be more actively managed through a multimodal rehabilitation programme in patients with Parkinson’s disease as well as their guardians.

## 2. Materials and Methods

### 2.1. Study Design and Ethical Approval

This study used a multicentre parallel randomised controlled, single-blind design. The study was conducted from February 2020 to August 2020, using outpatients at Hospitals A and B in Seoul, and Hospital C in Gyeonggi-do. The purpose, content, risk factors, and participation, which could be stopped at any time during the research without any disadvantage were explained to the participants and their guardians. This study was conducted in accordance with the Declaration of Helsinki, and approved by the Institutional Review Board (IRB) of Inje University (protocol code INJE 2019-11-028-001 and date, 29 January 2020).

### 2.2. Participants and Procedures

A total of 75 participants were recruited, of which 60 were randomly assigned to the experimental and control groups after meeting the inclusion criteria. The inclusion criteria were as follows: (1) those who were diagnosed with Parkinson’s disease by a neurologist using magnetic resonance imaging (MRI), and have been diagnosed more than six months before; (2) those in stage 3 or higher based on the Hoehn and Yahr staging scale [25], and (3) those who do not have cognitive or hearing/vision impairment with a Mini-Mental Status Examination (MMSE) score of 20 or higher, and can follow the instructions. Those with neurological diseases other than Parkinson’s disease, and those who cannot participate in treatment intervention due to medical conditions or other reasons were excluded.

All evaluations and interventions were performed by occupational therapists with more than three years of experience, and the principal investigator provided education on evaluation and intervention methods (Figure 1).

### 2.3. Intervention Method

#### 2.3.1. Multimodal Rehabilitation

The multimodal rehabilitation intervention, the therapy applied to the experimental group, was developed to include daily life training, home environment modification, fine motor exercises, fall prevention exercises, and guardian education based on the UK Parkinson’s Disease Occupational Therapy Guidelines [26] and the Dutch Parkinson’s Disease Occupational Therapy Guidelines [27]. Previous works were also included [28,29,30].

Daily life training consisted of general tips for successful daily life: cue methods applicable to training (using visual, auditory, and rhythmic cues); how to deal with dangerous situations in daily life; the introduction of assistive devices; and daily activities using assistive devices. Home environment modification consisted of a checklist of the home condition, and general and specific contents to be referred to when modifying the home environment. Fine muscle exercises consisted of warm-up exercises, manipulation of objects in the hands, and clay activities with the goal of functional use of the hands through improvement of hand coordination. Fall prevention exercises consisted of stretching and agility training. Guardian education consisted of education on training technology for patients, time management strategies, and routine management. The multimodal rehabilitation intervention was provided for 50 min per session, twice a week, for a total of 10 sessions. Table 1 shows the details of each session of the multimodal rehabilitation intervention.

#### 2.3.2. Traditional Rehabilitation

A traditional intervention was performed as a control intervention method. The traditional intervention consisted of task-oriented training, joint exercise, and daily living training.

Joint exercise consisted of passive joint exercise, active auxiliary joint exercise, active joint movement, and resistance joint exercise. Task-oriented training was performed step by step in small muscle activities, such as moving pegs and stacking blocks, as well as balance activities, such as standing up and moving various directional rings, and giving and receiving a ball on a balance board in consideration of the patient’s level of motor function. Daily life training consists of moving on a chair, bed, or floor, using the bathroom, eating, writing, and putting on and taking off clothes. The traditional intervention was conducted for 50 min per session (10 min of joint exercise, 20 min of task-oriented training, and 20 min of training for activities of daily living), twice a week, for a total of 10 sessions.

#### 2.3.3. Outcome Criteria

##### The New Activity Daily Living Questionnaire

The new activity daily living (ADL) questionnaire is a daily life behaviour scale created from the perspective of patients with Parkinson’s disease. A total of 20 items are evaluated, including walking, eating, sitting, and standing, taking the first step, crossing the street, climbing stairs, writing, moving to bed, using the toilet, bathing, talking, and moving objects. It is scored on a 6-point scale, where a score of 0 points indicates independent performance; a score of 1 point indicates slow but independent performance; a score of 2 points indicates performance with mild difficulty, not requiring the assistance of guardians or assistive devices; a score of 3 points indicates performance with moderate difficulty, requiring the assistance of guardians or assistive devices; a score of 4 points indicates performance with much difficulty, requiring the assistance of guardians or assistive devices; and a score of 5 points indicates the inability of performance. The new ADL questionnaire showed a high internal agreement with Cronbach’s α of 0.962–0.966, and the test–retest reliability *r* was 0.79 [31].

##### Parkinson’s Disease Questionnaire-39

The Parkinson’s Disease Qustionnaire-39 (PDQ-39) is a scale specifically designed to assess the quality of life in patients with Parkinson’s disease. This scale is scored on a 5-point scale, consisting of a total of 39 items in eight domains: ten items on movement; six items on activities of daily living; three items on social support; four items on cognitive function; and three items on communication. Patients can access each domain in relation to events that have occurred in the past month. All eight domains are to be converted into 100 points each, and the total score is calculated by adding up the scores of all the domains and dividing them into eight equal parts. A higher score indicates a lower quality of life [32]. PDQ-39 showed high internal consistency with Cronbach’s α = 0.94.

##### Zarit Burden Interview Korean

The Zari Burden Interview Korean (ZBI-K) was used to measure the level of burden of care felt by the guardians of patients with Parkinson’s disease. According to a study by Mosley [11], ZBI is the most frequently used scale in clinical settings when evaluating the burden of care of the guardians of patients with Parkinson’s disease. In this study, the Burden Interview scale developed by Zarit et al. [33], which was translated and adapted by Bae et al. [34], was used. The ZBI-K consists of a self-reported questionnaire with a total of 22 items on the burden of the guardians as individuals and caregivers. Each item is scored on a 5-point scale, from 0 point for “strongly disagree” and 4 points for “strongly agree”. A higher score indicates a higher burden of care. ZBI-K showed high internal consistency with a Cronbach’s α of 0.94.

### 2.4. Statistical Analysis

The significance level for statistical analysis was set to *p* < 0.05. Frequency analysis was conducted for the general characteristics of the participants through descriptive statistics. A paired sample *t*-test was performed to compare before and after intervention in each group. An independent sample *t*-test was performed to examine the effectiveness of the intervention between the two groups. After the intervention, Pearson’s correlation analysis was performed to investigate the correlation between activities of daily living, burden of care, and quality of life in the experimental group.

## 3. Results

### 3.1. Multimodal Rehabilitation Changes

Table 2 shows the activities of daily living, quality of life, and burden of care of guardians in the experimental group before and after intervention by test. The score of the new ADL questionnaire was 57.1 ± 2.59 before the intervention and 51.7 ± 2.55 after the intervention (*p* < 0.001). The score of PDQ-39 was 43.5 ± 4.85 before the intervention and 43.5 ± 4.85 after the intervention (*p* < 0.001). The score of ZBI-K was 45.8 ± 3.58 before the intervention and 37.7 ± 4.78 after the intervention (*p* < 0.001). Comparing the scores before and after the intervention, all tests showed a statistically significant difference (*p* < 0.05).

### 3.2. Traditional Rehabilitation Changes

Table 3 shows the activities of daily living, quality of life, and burden of care of guardians in the control group before and after intervention by test. The score of the new ADL questionnaire was 57.7 ± 3.16 and 54.1 ± 2.71 before and after the intervention, respectively (*p* < 0.001). The score of PDQ-39 was 43.4 ± 4.55 before the intervention and 39.6 ± 4.28 after the intervention (*p* < 0.001). Meanwhile, the score of ZBI-K was 46.6 ± 2.84 before the intervention and 46.6 ± 2.84 after the intervention (*p* < 0.001). Comparing the scores before and after the intervention, all tests showed a statistically significant difference (*p* < 0.001).

### 3.3. Comparison the Experimental and Control Groups

To compare the treatment effects of the experimental and control groups, we examined whether there was a significant difference in the scores of the new ADL questionnaire, PDQ-39, and ZBI-K. After the intervention, the score of the new ADL questionnaire changed by 5.3 ± 1.69 in the experimental group and 3.5 ± 1.36 in the control group, with a greater change observed in the experimental group (*p* < 0.01). After the intervention, the score of PDQ-39 changed by 7.5 ± 1.94 and 3.7 ± 0.96 in the experimental and control group, respectively, with a greater change observed in the experimental group (*p* < 0.001). After the intervention, the score of ZBI-K changed by 8.1 ± 3.11 in the experimental group and 2.2 ± 1.20 in the control group, with a greater change observed in the experimental group (*p* < 0.001) (Table 4).

### 3.4. Correlation between the Dependent Variable in the Experimental Group

Table 5 shows the correlation between the activities of daily living, quality of life, and burden of care of guardians in the experimental group. The PDQ-39 and ZBI-K showed a positive correlation (*r* = 0.578, *p* < 0.05). Concomitantly, the PDQ-39 and the new ADL questionnaire scores also showed a positive correlation (*r* = 0.330, *p* < 0.05).

## 4. Discussion

This study aimed to assess whether the application of a multimodal rehabilitation programme improved the daily life, quality of life, as well as ease the guardians’ burden of care for patients with Parkinson’s disease as well as their guardians.

The multimodal rehabilitation intervention applied in this study was developed with the aim of preventing loss of activity, ensuring patient-centred evaluation and intervention, setting goals in cooperation with patients and guardians, and applying various interventions to strengthen participation in daily life activities. Radder et al. [22] recommended a programme that included self-management, functional mobility, caregiver management, compensation strategy for activities of daily living, and home environment modification for patients with Parkinson’s disease. Deane et al. [35] recommended addressing the problems that guardians mainly complain about when designing interventions for patients with Parkinson’s disease. Jansa and Aragon [36] emphasized the importance of educating patients with Parkinson’s disease on how to adapt to daily life functions and personal lifestyle to improve their quality of life. Thus, patient-centred treatment should be implemented, and it is recommended to set treatment goals through sufficient consultation with patients and their guardians. Welsby, Berrigan, and Laver [37] recommend providing programmes for patients with Parkinson’s disease that simultaneously address motor functions and activities of daily living, as well as include meaningful activities that can be performed at home. The multimodal rehabilitation intervention applied in this study was constructed by adaptively reflecting the recommendations of previous studies.

Clarke et al. [38] reported that 381 patients with Parkinson’s disease who were treated with occupational therapy and physical therapy for three months showed improvement in their activities of daily living. Tickle-Degnen et al. [39] provided patients with Parkinson’s disease with a rehabilitation programme that included exercise, daily living activities, walking, and self-management for six weeks, reporting an improvement in the quality of life of patients. Sturkenboom et al. [23] provided patients with Parkinson’s disease with home-based occupational therapy programme that included goal setting, compensatory strategies education, and ten weeks of occupational performance skills training and reported an improvement in occupational performance skills in patients. The programmes implemented in previous studies were structured by the researchers to provide task-oriented exercise, strength and stretching exercises, gait and balance training, occupational performance skill training and fine muscle exercise.

Unlike in previous studies, the multimodal rehabilitation programme provided in this study consisted of daily living training, guardian education, home environment modification, fine muscle exercise, balance training, and training using auxiliary tools with the aim of providing a patient-centred approach to patients with Parkinson’s disease. It has been emphasised that the ability to perform activities of daily living, cognitive function, and quality of life of patients could be improved by participating in meaningful tasks and engaging in valuable activities of daily living [40,41]. Thus, this study differed from previous studies in that it included programmes for activities of daily living, guardian education, and home environment modification. In this study, the experimental group improved more than the control group through improving the functional independence of patients, enabling them to safely perform activities of daily living, and reducing the burden of care of guardians with reduced care time due to less dependence of patients, as well as allowing patients and guardians to select the parts of the programme they needed.

In this study, the relationship between ZBI-K and PDQ-39 showed the highest correlation (*r* = 0.59). This indicates that the quality of life in patients improves as the burden of care is reduced. According to a study by Rajiah et al. [42], stigma and emotional stability in the quality of life of patients with Parkinson’s disease affected the burden of care of guardians. Tessitore et al. [43] reported that the quality of life in patients with Parkinson’s disease often affected the burden of care of guardians in the long-term management of the patients. Macchi et al. [44] reported that the higher the care burden of Parkinson’s disease patients, the more the adverse outcomes. In other words, the results of this study support that the lower the burden of care, the more positive the results. Moreover, a study by Miyashita et al. [11] showed that the quality of life in patients with Parkinson’s disease improved with higher social support from their families, thus lowering the burden of care. This was consistent with the results of our study, which showed that the burden of care of guardians decreased with an improvement in the quality of life in patients with Parkinson’s disease. There was a significant improvement in emotional stability and social support, which are areas of quality of life, and which may have greatly affected the reduction of the burden felt by guardians.

The relationship between the new ADL questionnaire and PDQ-39 in this study showed a positive correlation (*r* = 0.33). This indicates that the quality of life in patients improves as their level of independence in activities of daily living increases. In a study by Lawrence et al. [45] on the quality of life and depression of patients with Parkinson’s disease, activities of daily living did not affect the relationship between quality of life and depression; however, there was a statistically significant correlation between activities of daily living and quality of life. Kleiner-Fisman, Stern, and Fisman [46] reported that there was a similar relationship between the quality of life and activities of daily living in patients with Parkinson’s disease, where an impaired performance of daily living activities was highly associated with a degradation of the quality of life. Reuther et al. [47] found that the most unfavourable aspect for patients with Parkinson’s disease was limited participation due to reduced functional mobility and ability to perform activities of daily living, which were closely related to the quality of life. He et al. [48] verified the mediating effect of a decrease in ADL, which leads to an increase in the level of depression, and a decrease in quality of life. Depression was not measured in this study, but it was confirmed that ADL was related to quality of life. In line with the results of the previous studies, this study also showed that the burden of care of guardians was reduced with an improvement in independence in the activities of daily living of patients with Parkinson’s disease. These improvements were shown mostly in transfer and mobility factors, which were related to an improvement in the mobility and activities of daily living factors in the quality of life of the patients. These improvements were expected to affect the emotional stability, and thus improve the quality of life of the patients.

As for patients with Parkinson’s disease, treatment is often delayed due to misconceptions about the disease, and although drug treatment is effective, it can cause a considerable economic burden due to adverse effects associated with the drugs [49]. To prevent such adverse effects, non-pharmacologic rehabilitation is performed; however, the focus is not on patients with Parkinson’s disease [50]. Based on our results, active implementation of a multimodal rehabilitation programme for patients with Parkinson’s disease can help improve the patient’s function, and thus prevent falls and increase their independence in performing activities of daily living. Moreover, rehabilitation can prevent problems caused by motor symptoms that may occur later, thereby reducing cost burdens on patients.

This study has certain limitations. Due to a small sample size, the results are not conclusive across all patients with Parkinson’s disease as we only included those who were at stage 3 on the Hoehn and Yahr Scale. In addition, there has been no follow-up evaluation at three or six months after the intervention, making it difficult to confirm whether the effect of the intervention is long-term. In future research, it is necessary to observe the effect of the programme across different stages of the Hoehn and Yahr Scale, as well as to observe the continuity of the effect of the treatment programme through a three-month evaluation post- intervention. Furthermore, while there are many facilities, such as dementia relief centres for dementia patients, facilities for Parkinson’s disease patients are currently limited. Therefore, future research on programmes to professionally manage patients with Parkinson’s disease by proving the effectiveness of multimodal rehabilitation in the community rather than in a hospital setting is necessary.

## 5. Conclusions

Multimodal rehabilitation intervention for Parkinson’s disease is more effective in improving the activities of daily living and quality of life of patients, as well as easing the burden of guardians, as compared to the traditional interventions. Thus, we suggest applying a multimodal rehabilitation intervention to improve activities of daily living and quality of life of patients with Parkinson’s disease, as well as reduce the burden of care of their guardians.

## Figures and Tables

**Figure 1 healthcare-10-01888-f001:**
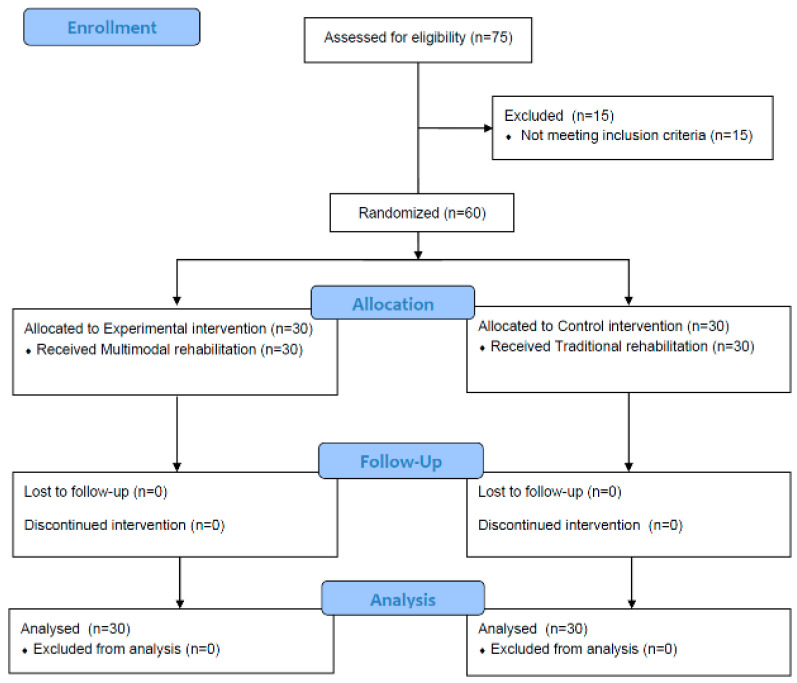
Study design.

**Table 1 healthcare-10-01888-t001:** Multimodal Rehabilitation.

Session	Multimodal Rehabilitation
1A. Fine motor exercise (30 min)	∗ Preliminary exercise (10 min): finger tapping, thumb opposition, finger abduction & adduction, selective movement ∗ In-hand manipulation training (5 min) ∗ Clay activity (15 min): making ball shape, remove the peg in the clay, make clay long
1B. Education (20 min)	∗ Home environment modification education ∗ Caregiver education: training skill for client, time management strategy, daily management
2A. Fine motor exercise (30 min)	∗ Same as 1A Session
2B. ADL training (20 min)	∗ Focus on what the client needs most ∗ Training using assistive device ∗ The caregiver observes the treatment and provides training on the cueing method provided to the client.
3~4A. Fall prevention exercise (30 min)	∗ Stretching exercise (10 min) ∗ Agility training (20 min): ladder training
3~4B. ADL training (20 min)	∗ Same as 2B Session
5. Visiting home & Feedback	∗ Provide feedback after checking home environment modification by visiting the home
6~9A. Fall prevention exercise or Fine motor exercise(30 min)	∗ Stretching exercise or Preliminary exercise ∗ Agility training or in-hand manipulation training, clay activity
6~9B. ADL training (20 min)	∗ Focusing on ADL that were difficult when performed at home ∗ Occupational performance for caregivers, problems solving
10. Visiting home & Feedback	∗ Provide feedback after visiting the home to check the actual patient’s daily life performance

**Table 2 healthcare-10-01888-t002:** Multimodal Rehabilitation Changes.

Items	M ± SD		*t*	*p*	Within Group Change 95% CI
	Pre-Test	Post-Test			
The new ADL questionnaire	57.1 ± 2.59	51.7 ± 2.55	16.076	0.000 ***	5.4 (4.66, 6.03)
PDQ-39	43.5 ± 4.85	36.0 ± 3.82	19.888	0.000 ***	7.5 (6.79, 8.36)
ZBI-K	45.8 ± 3.58	37.7 ± 4.78	13.241	0.000 ***	8.1 (6.82, 9.33)

*** *p* < 0.001. ADL, activity daily living; PDQ, Parkinson’s disease questionnaire; ZBI-K, Zarit burden interview Korean.

**Table 3 healthcare-10-01888-t003:** Traditional Rehabilitation Changes.

Items	M ± SD		*t*	*p*	Within Group Change 95% CI
	Pre-Test	Post-Test			
The new ADL questionnaire	57.7 ± 3.16	54.1 ± 2.71	13.235	0.000 ***	3.5 (2.98, 4.08)
PDQ-39	43.4 ± 4.55	39.6 ± 4.28	19.784	0.000 ***	3.7 (3.34, 4.11)
ZBI-K	46.6 ± 2.84	44.4 ± 2.36	9.311	0.000 ***	2.2 (1.70, 2.67)

*** *p* < 0.001. ADL, activity daily living; PDQ, Parkinson’s disease questionnaire; ZBI-K, Zarit burden interview Korean.

**Table 4 healthcare-10-01888-t004:** Comparison of the experimental and control groups.

Items	Experimental Group	Control Group	*t*	*p*	Between Group Change (95% CI)
The new ADL questionnaire	5.3 ± 1.69	3.5 ± 1.36	−4.236	0.001 **	1.8 (0.95, 2.66)
PDQ-39	7.5 ± 1.94	3.7 ± 0.96	−9.048	0.000 ***	3.8 (2.99, 4.70)
ZBI-K	8.1 ± 3.11	2.2 ± 1.20	−9.000	0.000 ***	5.9 (4.55, 7.20)

** *p* < 0.01, *** *p* < 0.001. ADL, activity daily living; PDQ, Parkinson’s disease questionnaire; ZBI-K, Zarit burden interview Korean.

**Table 5 healthcare-10-01888-t005:** Correlation between the dependent variable in the experimental group.

	The New ADL Questionnaire	ZBI-K
ZBI-K	0.259	
PDQ-39	0.330 *	0.578 *

* *p* < 0.05. ADL, activity daily living; PDQ, Parkinson’s disease questionnaire; ZBI-K, Zarit burden interview Korean.

## Data Availability

All the data are available in this paper.

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
