# Peer review of "Effects of Multimodal Rehabilitation on the Activities of Daily Living, Quality of Life, and Burden of Care for Patients with Parkinson’s Disease: A Randomized Control Study"

_healthcare, 2022, doi:10.3390/healthcare10101888_

Round 1
Reviewer 1 Report
Dear Authors,
It is a simple but extremely interesting study within a very relevant and current context. It is well written and methodologically well presented. It's pleasant to read. Congratulations to the authors!
I only had a single doubt in the expression: “sluggishness as compared with tremors and stiffness,”… used in line 38 that I would like authors to be able to review.
Author Response
Deer Reviewer
Thank you for your review.
The sluggishness is expressed as bradykinesia, which is a more accurate expression.
Reviewer 2 Report
The article "Effects of multimodal rehabilitation on the activities of daily living, quality of life, and burden of care for patients with Parkinson's disease: A randomized control study" presents a valuable study of a modified rehabilitation program aiming to focus on a patient centered model of rehabilitation. The introduction provides excellent study background and rationale. The randomized study design is well structured and the study eligibility criteria are properly selected in consideration with the endpoints and expected outcomes. The bias in patient reported outcomes is planned to be eliminated through single blinded design. Now I am not entirely convinced that single-blindness is fully achievable considering that the patients at this stage of the disease most probably already have exposure to rehabilitation programmes. Authors do not mention whether the selected participants had already taken part in any rehabilitation programme. Such prior experience and/or preexisting knowledge of the nature of rehabilitation programmes could allow the participants to tell if they are in the control group or in the experimental group, hence the bias. Moreover, during the informed consent process, investigators should have provided certain information regarding the nature of the study and the traditional and the tested programme. This bias could be eliminated to some extent if the study excludes participants who have already participated in any rehabilitation program but again this does not guarantee that the patients would not have knowledge on what a traditional rehabilitation programme consists of. In my opinion, although randomized blinded studies are at the top of the evidence pyramid and authors' desire to use such design is understandable, it is not necessarily needed to confirm the hypothesis. A cross-over randomized non-blinded design could allow for a certain reduction of patient bias and could be used in further research instead of a single blinded design. Worthy of special acknowledgement is the study limitations section as it demonstrates authors' quality of work (this is not always the case). The limitations outline the future areas of research making this relatively small study an excellent starting point for the development of a much needed more patient-centric approach.This section could include a note regarding the single blinded design I mention above however I would leave this judgement to the editor and would recommend this article for publication as is. The chosen methods are well selected and described in the article. Analysis is very well presented. Conclusions reflect very well the results. The article is very well written and easy to read. No language corrections or proofreading is not necessary. I have not detected plagiarism or inproper citing. I am happy to recommend this article for acceptance.Author Response
Deer Reviewer
Thank you for your review.
We will consider a cross-over randomized non-blinded design in future studies.
Reviewer 3 Report
Dear authors,
The study they present is very interesting for the scientific community and Parkinson's disease rehabilitation clinical teams.
However, they need to make some adjustments, modifications and improvements:
Abstract
The abstract does not meet the criteria established in the CONSORT guidelines for clinical trials.
Authors should structure the abstract in trial design, methods, results and conclusions.
I suggest reading carefully the following bibliographic references for the structure and writing of the abstract.
Hopewell S, Clarke M, Moher D, Wager E, Middleton P, Altman DG, et al. CONSORT for reporting randomized controlled trials in journal and conference abstracts: explanation and elaboration. PLoS Med 2008;5:e20.
Hopewell S, Clarke M, Moher D, Wager E, Middleton P, Altman DG, et al. CONSORT for reporting randomised trials in journal and conference abstracts. Lancet 2008;371:281-3.
Introduction
Improve the presentation of the general objective of the study (lines 94-99).
Methods
I suggest reviewing the CONSORT guidelines for the publication of clinical trials.
In point 2.1. Design of the study, the intervention of the control group and the experimental group must appear, in addition to compliance with the CONSORT guidelines.
It is mandatory to mention the ethics committee and indicate the approved protocol code, in addition to compliance with the Declaration of Helsinki and acceptance of informed consent.
Randomization data must also be shown.
Figure 1.- It uses the CONSORT flowchart.
Lines 106-110 must appear in item 2.4.
Results
The incorporation of confidence intervals is necessary.
Discussion
It is in deficit.
I suggest comparing the quantity results found in this study with similar studies.
Kind regards,
Author Response
Deer Reviewer
Thank you for your review.
Abstract
We revised according to the CONSORT guidelines.
Introduction
Corrected the expression of objective of the study.
Methods
We revised according to the CONSORT guidelines.
The description of the experimental group control intervention is detailed in Sections 2.3.1 and 2.3.2.
Indicated compliance with the Declaration of Helsinki and approved protocol code of the ethics committee.
Figure 1 used the CONSORT flowchart.
Results
Confidence intervals are described.
Discussion
We improved the comparative analysis with other studies.
Round 2
Reviewer 3 Report
Dear authors,
The article includes major improvements.
However, you continue to fail to comply with CONSORT international guidelines in the abstract.
Figure 1 is of poor quality. Please include it in adequate size and quality.
The introduction continues to be very brief, as is the discussion. I suggest going deeper in both sections.
Kind regards,
Author Response
We made revisions based on the reviewers' comments.
Abstract
Revised according to the CONSORT guidelines.
Introduction
Corrected the expression of objective of the study and added an introduction.
Methods
Revised according to the CONSORT guidelines.
Indicated compliance with the Declaration of Helsinki and approved protocol code of the ethics committee.
Figure 1 used the CONSORT flowchart.
Results
Confidence intervals are described.
Discussion
Improved the comparative analysis with other studies.